# Comparison of Properties with Relevance for the Automotive Sector in Mechanically Recycled and Virgin Polypropylene

**Abdelhak Ladhari** [1],[†] , **Esra Kucukpinar** [2] , **Henning Stoll** [1] **and Sven Sängerlaub** [1],[3],[*],[†]

1    Department of Mechanical, Automotive and Aeronautical Engineering,
Munich University of Applied Sciences HM, Lothstraße 38, 80335 Munich, Germany;
ladhari.abdelhak@hm.edu (A.L.); henning.stoll@hm.edu (H.S.)
2    Fraunhofer Institute for Process Engineering and Packaging IVV, Giggenhauser Strasse 35,
85354 Freising, Germany; esra.kucukpinar@ivv.fraunhofer.de
3    Department of Building Services Engineering, Paper and Packaging Technology and Print and Media Technology,
Munich University of Applied Sciences HM, Lothstraße 38, 80335 Munich, Germany
*    Correspondence: sven.saengerlaub@hm.edu
†    Both authors contributed equally to this work.

**Abstract:** Polypropylene (PP) has a high recycling potential. However, the properties of mechanically recycled PP (R-PP) have not been fully compared to those of virgin PP (V-PP). Therefore, in this study, properties of R-PP and V-PP were compared using data from recyclers, virgin plastic suppliers, and the literature. The properties of recyclates could not be directly correlated either with the properties of the virgin polymers from which the recyclates were made or the recycling parameters. It was found that the MFR of R-PP was higher; MFR R-PP had a median value (m) of 11 g/10 min while MFR V-PP had a median value of 6.3 g/10 min (at 230 °C and with 2.16 kg). In terms of mechanical properties, in many cases R-PP exhibited stiffer and more brittle behavior, with a slightly higher Young's modulus ($E_{\text{R-PP}}$ = 1400 and $E_{\text{V-PP}}$ = 1200 MPa), a reduced elongation at break ($\mathcal{E}b_{\text{R-PP}}$ = 4 l.-% and $\mathcal{E}b_{\text{V-PP}}$ = 83 l.-%), and notched charpy impact strength ($NCIS_{\text{R-PP}}$ = 4.8 and $NCIS_{\text{V-PP}}$ = 7.5 kJ/m$^2$). However, the values for every property had a broad distribution. In addition to existing information from the literature, our research sheds fresh light on the variation of the characteristics of recycled polypropylenes presently on the market.

**Keywords:** recycling; polypropylene; polypropylene recyclates; mechanical properties; rheological properties; automotive





## 1. Introduction

### 1.1. Plastics in the Car Industry

Plastics and the automobile industry have a shared history dating back to 1839, when Charles Goodyear developed vulcanized rubber by altering the mechanical characteristics of natural rubber from the Pará rubber tree. This rubber was the one of the world's first polymers; it had a high relevance for cars and was quickly adopted for use in automobile tires [1].

After World War II, the widespread usage of plastics in the automotive sector underwent a major shift. The potential to create a low-cost petroleum-derived gasoline provided a cohesive and dependable raw material for the manufacturing of low-cost plastics [2]. This presented the automobile sector with many possibilities. Today, the automotive industry ranks third in terms of plastics consumption, after packaging and building and construction [3].

The use of plastics in the car manufacturing industry presents several advantages, such as the following: (1) Fuel efficiency: The incorporation of plastics into vehicle design improves fuel efficiency and reduces overall emissions. Substituting plastics for heavier materials results in an overall weight decrease, with a 10% weight reduction resulting in a 3% to 7% increase in fuel economy [4]. (2) Design and innovation: External plastic vehicle cladding enables automobile designers to develop novel designs that would not

be achievable with metal. (3) Safety: Polymers are used to construct automobile safety systems such as airbags and seatbelts. Exterior items, for example bumpers, serve as shields, assisting in the prevention and/or mitigation of damage to the vehicle's body caused by low-speed accidents. Bumpers are important because they lower the chance of pedestrian harm in low-speed crashes by absorbing energy and separating pedestrians from the more rigid structural sections of the automobile [5]. (4) Weather resistance: synthetic coatings on metal surfaces are employed to minimize the risk of corrosion caused by salt damage, high heat, and water exposure.

The percentages of the top six plastics and polymers used in automobiles with different powertrain configurations manufactured between 2016 and 2020 are as follows: 44% PP, 4% PBT, PET, 5% ABS, 7% PE, 8% PA, 9% PUR, and 23% others [6]. There are currently approximately 39 different types of plastics and polymers used in the manufacture of automobiles [7]; however, 72 kg of the 158 kg of plastic used is polypropylene (PP) [6]. European manufacturers alone use over one million tons of PP per year [8], which gives PP a huge recycling potential. Key automotive application features of PP are its low coefficient of linear thermal expansion and specific gravity, high chemical resistance, and good weatherability, processability, and impact/stiffness balance. PP characteristics may be further improved with the use of additives such as long-glass fiber reinforcing. Main applications include bumpers, air ducts, battery cases and trays, fender liners, interior trim, instrumental panels, and door trims [6]. Typical PP property requirements for automotive applications are given in Table S1.

Automotive manufacturers worldwide have committed to reducing their industrial footprints and plastics have been identified as a critical component of these environmental sustainability initiatives. Volvo, the Swedish car manufacturer, stated that by 2025, at least 25% of the plastics used in its vehicles would come from recycled materials. As recyclates are increasingly being used to create new automotive components [9], competition between the recycled plastics may occur. To prevent production problems, it is necessary to ensure a continuous supply of recycled material throughout the manufacturing process of an automobile component. Therefore, recycling becomes interesting, but in order to use recycled plastics as a substitute, recyclers must guarantee that they have sufficient technical and quality properties compared to the virgin material. The substitution factor is the key parameter that describes the effect of substituting recycled plastic for virgin plastic, it being the ratio of virgin to recycled material for the same application. A substitution factor of 1 indicates that recycled material can replace virgin plastic without increasing the product's weight. When more recycling material is required to compensate for physical properties, the substitution factor is less than 1. If substitution of virgin materials in products is not possible and the R-PP is instead used in lower-quality products, it is called "downcycling" [10].

### 1.2. Degradation of PP during Multiple Extrusions

During extrusion, PP degrades, which is a known phenomenon. Different research groups have conducted multiple extrusion experiments to analyze the influence of this degradation. The subjects of these investigations were thermal properties (DSC, TGA), rheological measurements (MFR), mechanical properties, and spectroscopy [11–33]. Molecular weight decreases [13,34] by chain scission [28,35] causing lower viscosity and higher MFR [13,15,25,36–38]. The molecular weight distribution is narrowed, which is associated with higher flowability of the polymer melt but also lower mechanical performance of parts thereof [34]. A higher extrusion temperature results in a higher MFR [37] caused by more chain scission. In addition, a higher crystallinity [11,15,36] can be explained by smaller macromolecules, which arrange and crystallize easier. A higher crystallinity due to chain scission during multiple extrusion has been associated in several studies (but not all) with a higher tensile modulus and tensile strength [36] but a lower elongation at break [11,16]. However, in contrast, a lower tensile strength (here yield stress) was

observed with more extrusion cycles [13]. Virgin PP has a higher resistance to break due to its higher molecular weight [30].

PP recyclates often contain traces of impurities such as other polymers (PA, PE, PET) [39,40]. In such samples, lower tensile impact strength and strain but higher tensile strength and tensile modulus were observed [39].

### 1.3. Intention of This Study

The aim of this study was to compare the mechanical and rheological properties of market-available recycled polypropylene (R-PP) with those of virgin polypropylene (V-PP). We used the presented results of product data sheets and of various studies. Our hypotheses were as follows:

**Hypothesis 1 (H1).** *R-PP has a higher MFR, tensile modulus, and yield strength compared to V-PP;*

**Hypothesis 2 (H2).** *R-PP has a lower elongation at break and charpy impact strength compared to V-PP.*

We based our assumption on the phenomenon of chain scission, which causes a higher macromolecule mobility and therefore a higher crystallinity and lower elongation at break.

## 2. Results and Discussion

### 2.1. Rheological Properties (MFR)

The melt flow rate (MFR) is a parameter that indicates the resistance to flow of a polymer melt (viscosity) at a particular temperature and applied force over a specified length of time. All values here were measured at 230 °C and with 2.16 kg weight. Full viscosity curves were not provided by the suppliers.

Figure 1 shows that both R-PP and V-PP MFR values spanned a broad range. $\text{MFR}_{\text{R-PP}}$ had a median value of 11 g/10 min (this means that 50% of the values collected for R-PP were over 11 g/10 min) while $\text{MFR}_{\text{V-PP}}$ had a median value of 6.25 g/10 min.

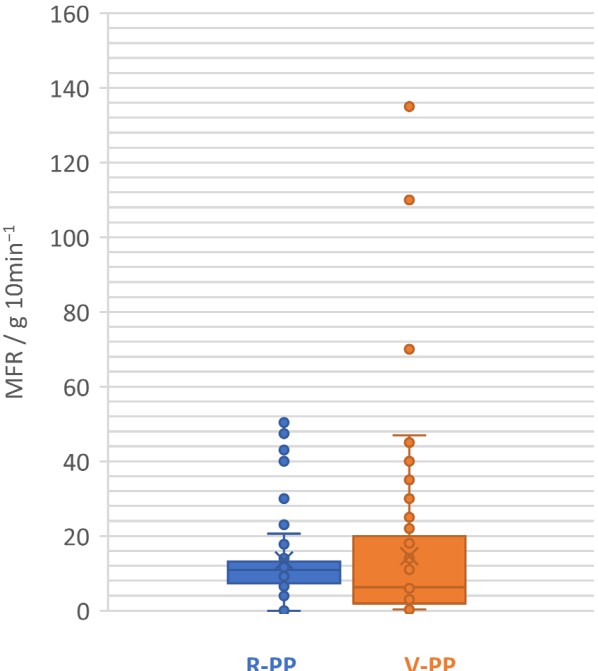

**Figure 1.** MFR values of recycled (R-PP) and virgin polypropylene (V-PP) at 230 °C and with 2.16 kg; for data, see Table S2.

MFR indirectly correlates with molecular weight ($M_w$), with a low melt flow rate indicating a high $M_w$ [41]. Multiple extrusions have been demonstrated to result in a reduction in the average molecular weight [24,42]. The decrease in molecular weight has been ascribed to a chain scission mechanism caused by thermomechanical effects [43]. Therefore, the result matches prior findings on the $M_w$ reduction of PP due to the recycling process. Stabilizers may be used to avoid this increase in MFI, but they also are exhausted through recycling operations [15]. Information about their use in the present samples is unknown.

The values found for R-PP were still acceptable both for injection and blow molding processes. MFR values of 1 to 100 g/10 min (230 °C/2.16 kg) for injection molding, 1.0–2.5 g/10 min for extrusion blow molding, and 10–25 g/10 min for injection blow molding have been reported to be suitable [34]. Injection molding and the blow molding technique are extensively utilized in the automobile sector due to their ability to produce complicated products in a variety of forms and shapes that would be difficult to achieve with other technologies even at a considerably greater cost [44].

### 2.2. Mechanical Properties

#### 2.2.1. Tensile Modulus

R-PP had slightly a higher tensile modulus than V-PP (Figure 2); this could be explained by the higher crystallinity of the R-PP. With each recycling cycle, the degree of crystallinity increases [16]. As molecular weight decreases, chain mobility increases and crystallization happens in a more organized manner [45]. Crystalline materials are considerably stiffer than the same plastics in an amorphous state [15,46] due to a higher tensile modulus.

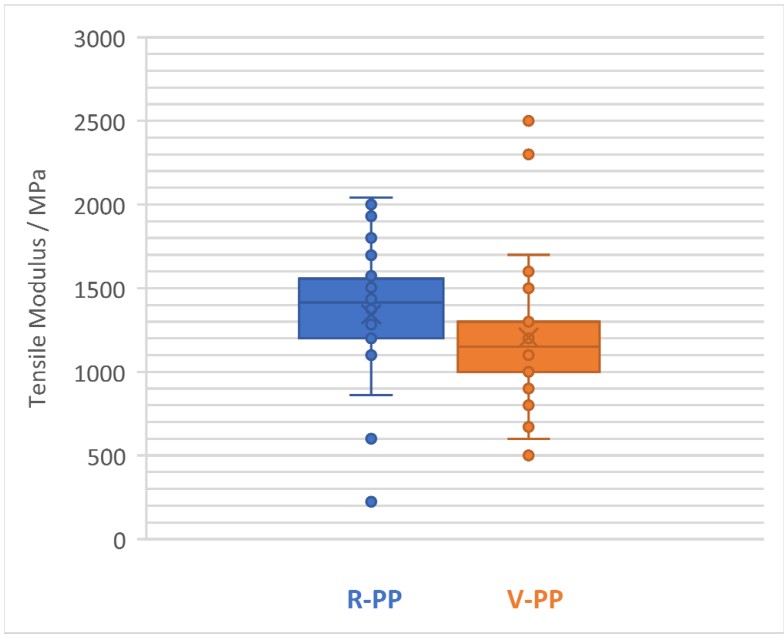

**Figure 2.** Tensile modulus values of R-PP and V-PP; for data, see Table S2.

A higher tensile modulus can also be caused by inorganic fillers used to reinforce PP. Fillers can act as nucleating agents. Nucleating agents increase crystallinity [47–49].

#### 2.2.2. Yield Strength

The median of the R-PP yield strength ($\sigma_y$) was similar to that of the V-PP (Figure 3). However, the R-PP values had a wider distribution. The possible increase in yield strength could be associated with the higher crystallinity of the recycled polypropylene, which results from increased intermolecular bonding in the crystalline phase [45].

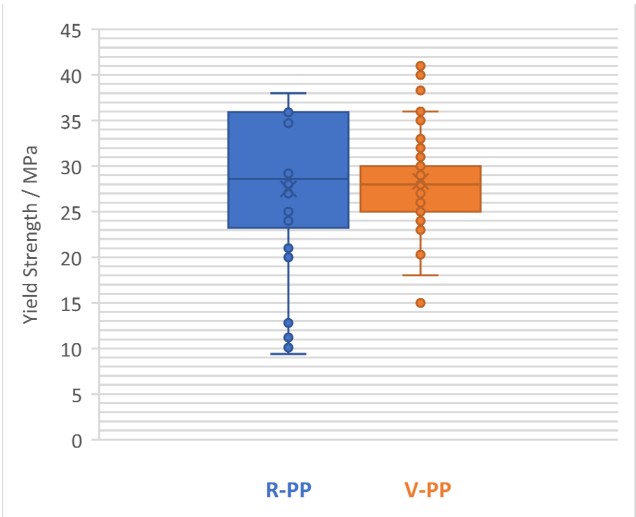

**Figure 3.** Yield strength of R-PP and V-PP; for data, see Table S2.

### 2.2.3. Elongation at Break

The median of the elongation of break of R-PP was lower than that of V-PP (Figure 4). The elongation at break is expected to drop as the molecular weight decreases, because shorter chains are easier to untangle, allowing the test specimen to break at a lower elongation percentage. Along with variations in the molecular weight, the increase of crystallinity as a consequence of chain scission also influences the elongation at break. In contrast to tensile modulus, increasing crystallinity decreases the value of elongation at break because the stiff crystal particles are unable to stretch like the amorphous polymer chains [50]. Another explanation for the reduced elongation of break of recyclates is impurities [40]. Their contents in the recyclates were not known and their impact could therefore not be evaluated.

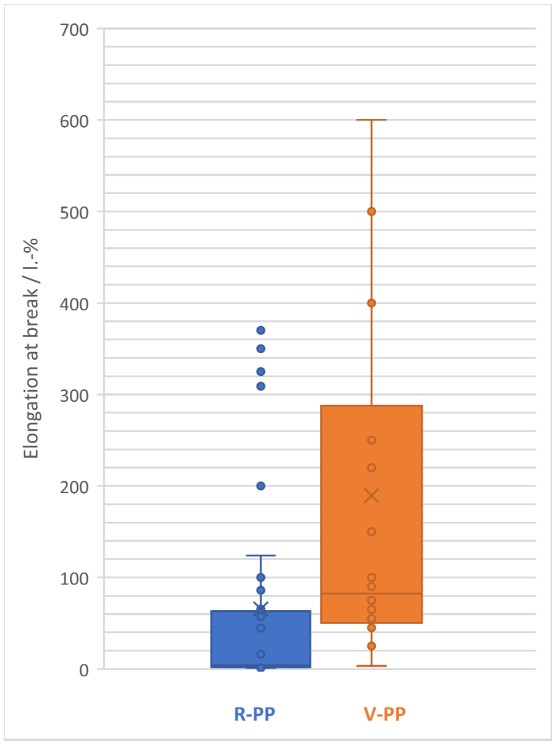

**Figure 4.** Elongation at break of R-PP and V-PP; for data, see Table S2.

### 2.2.4. Charpy Impact Strength

Figure 5 shows a significant decrease in the median of the charpy impact strength for R-PP in comparison to V-PP. In general, a high molecular weight [41] and a narrow molecular weight distribution enhance impact resistance, whereas increasing crystallinity and voids reduce it. The lower charpy impact strength makes R-PP unsuitable for use in high concentrations in new car bumpers, because the energy-absorbing medium in a good bumper system should withstand the impact and absorb the crash energy to limit vehicle damage. A good bumper system should also have enough resilience and deformability [5].

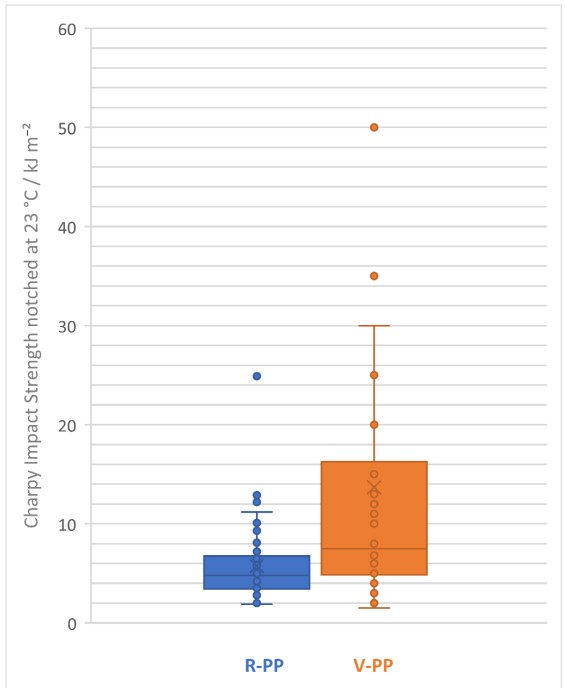

**Figure 5.** Charpy impact strength of R-PP and V-PP; for data, see Table S2.

### 2.2.5. Correlation of Properties

In Figure 6, tensile modulus and MFR are correlated, showing data specifications for both results. There was no correlation observed between the tensile modulus and MFR. However, values for V-PP scattered more than those for R-PP. Due to the fact the exact compositions of the PPs were unknown, results could not be significantly correlated.

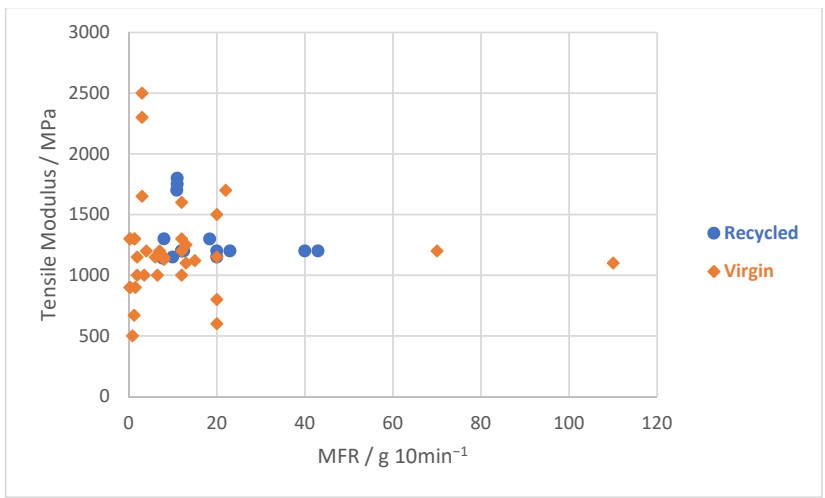

**Figure 6.** Tensile modulus and MFR values of R-PP and V-PP.

In Figure 7, tensile modulus and elongation at break are correlated, showing data specifications for both results. A higher crystallinity correlated with a higher tensile strength and a lower elongation at break. There was no strong correlation observed between the tensile modulus and elongation at break. However, values for R-PP scattered more than those for V-PP. Due to the fact the exact compositions of the PPs were unknown, results could not be significantly correlated.

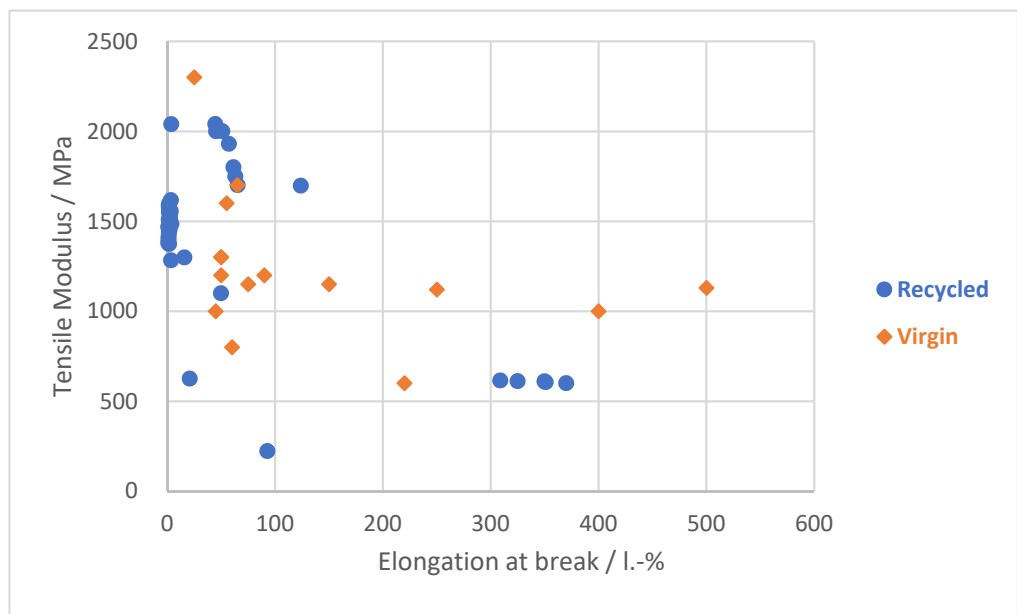

**Figure 7.** Tensile modulus and elongation at break values of R-PP and V-PP.

## 3. Materials and Methods

In order to gather information on the mechanical and rheological properties of R-PP, over 570 plastic recycling plants' websites were visited or directly contacted through email to get information in the primary research stage. Three percent of the companies (i.e., 18 plants) had datasheets on their websites or were willing to provide them upon request. In the secondary research, data were mainly collected from research papers and journal articles. For V-PP, most of the information was obtained from the websites of plastic suppliers, where most suppliers provided product datasheets.

To compare the technical properties of R-PP with those of V-PP, quantitative data were required. Table 1 displays the data collection matrix.

**Table 1.** Data collection matrix (see File S1); databases accessed March–July 2021.

| Perspective | Primary Research | Secondary Research |
|---|---|---|
| R-PP | Datasheets from plastic recycling plants | Research papers, journal articles |
| V-PP | Datasheets from plastic suppliers | Reference books, publications |

Table 2 summarizes the data gathered on V-PP and R-PP, as well as the equivalent number of values collected.

**Table 2.** Data and number of values collected.

| Property | R-PP | V-PP |
|---|---|---|
| Melt flow rate (MFR) | 51 | 74 |
| Tensile modulus ($E$) | 53 | 37 |
| Tensile strength at yield ($\sigma_y$) | 40 | 61 |
| Elongation at break ($\mathcal{E}_b$) | 45 | 20 |
| Notched charpy impact strength | 55 | 48 |

The results of the data collection are given in Table S2. The data for V-PP lacked information on the molecular weight distribution, conformation, crystallinity, extrusion process, flexural strength, additives, and the material's chemical composition. The data for R-PP lacked, in addition, information about the number of extrusions and impurities. There was also no information about the grades of PP (homo-polymer, random copolymer, or block copolymer). These three grades cannot be separated during the recycling sorting process because the majority of plastic recycling facilities sort plastics using near-infrared reflection (NIR) spectroscopy [24]; this sorting method can sort the main polymers but not the grades of the polymers. All types of PP are blended and marketed as a single product after recycling. To obtain insight into the rheological and mechanical characteristics of various kinds of V-PP, datasheets were randomly chosen.

There are two widely used testing methods for determining the tensile properties of resin samples: (1) American Society of Testing Materials (ASTM): the ASTM D638 standard is primarily used in North America. (2) International Standards Organization (ISO): the ISO 527 standard is used primarily throughout Europe and Asia. While there are many variations between the injection molding and analytical testing techniques specified by the ISO and ASTM standards [42], the findings achieved using each approach are similar for the majority of tensile tests [43]. Therefore, the outcomes of both methods were evaluated together. However, there was a considerable variation in the impact resistance recorded for various resin types using each protocol [15] (ASTM D6110 and ISO 179); thus, only results from ISO 179 were compared in this study. As for MFI, the two major test standards are ASTM D1238 and ISO 1133; both standards measure the same property and should deliver the same result when properly conducted.

To make comparisons easier and to gain a better understanding of the distribution of the data collected for each property of V-PP and R-PP, Excel box plots were used. In a box plot, quantitative data are split into quartiles, with a line drawn between the first and third quartiles to represent the median. Outside the first and third quartiles, the minima and maxima are depicted with lines. The interquartile range (IQR) is the gap between the third and first quartiles. Excel classifies a data item as an "outlier" if it is greater than the third quartile by 1.5 times the IQR or smaller than the first quartile by 1.5 times the IQR.

## 4. Conclusions

As expected from and in harmony with the general theory of polymer processing and polymer physics, the findings of our study indicate that the median of market-available recycled polypropylene is lower for the impact strength and the elongation at break but slightly higher for the tensile modulus compared to market-available virgin polypropylene.

Even though it was impossible from the available market data to draw a direct dependency between the properties of virgin and recycled PP, it can be assumed that both correlate with each other (see Section 1.2). It can be assumed further that recycling causes changes in the mechanical properties. Interestingly, and as expected, the properties of virgin PPs scattered widely, reflecting the many PP grades available on the market.

Our results should be considered when making substitutions of virgin material in car products. The value of the recyclates generated from PP-based automotive parts can be increased by identifying applications in which the recyclates' characteristics of a higher tensile modulus can be uniquely utilized even though the recycled material is more brittle with a lower elongation at break. It is worth mentioning that, because of the high variability of results, cases can be found where recycled PP has higher values than virgin PP for a technical parameter.

The exact composition, molecular weight distribution, and conformity of the R-PP and V-PP samples were not known. Therefore, it was not possible in this study to attribute properties and property changes to additives or, in the case of R-PP, to possible impurities.

**Supplementary Materials:** The following are available online at https://www.mdpi.com/article/10.3390/recycling6040076/s1. Table S1: Key functional properties of PP for automotive application, Table S2: Data collected from product data sheets, R: recycled PP, V: virgin PP; PP types and compositions unknown, File S1: References for polymer properties.

**Author Contributions:** Conceptualization, A.L., S.S. and H.S.; methodology, A.L.; formal analysis, A.L.; investigation, A.L.; data curation, A.L.; writing—original draft preparation, A.L., S.S.; writing—review and editing, S.S., E.K. and H.S.; supervision, S.S.; funding acquisition, S.S. All authors have read and agreed to the published version of the manuscript.

**Funding:** This work was financially supported by the Munich University of Applied Sciences HM and the German Research Foundation (DFG) through the "Open Access Publishing" program.

**Institutional Review Board Statement:** Not applicable.

**Informed Consent Statement:** Not applicable.

**Data Availability Statement:** All data sources are listed in the references.

**Conflicts of Interest:** The authors declare no conflict of interest.

**Remark:** Results of this work were used previously for the Bachelor thesis of the main author Abdelhak Ladhari.

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
