# Peer review of "Comparison of Properties with Relevance for the Automotive Sector in Mechanically Recycled and Virgin Polypropylene"

_recycling, doi:10.3390/recycling6040076_

Round 1

Reviewer 1 Report

1) Different grades of polypropylene, due to the crystallinity of the structure of the difference, its own performance range is very large. Different recycled polypropylene may get different results and conclusions from the paper. The polypropylene and recycled polypropylene described in the paper are to be used with a strictly specified source and scope of collection.

2)The conclusions presented in this paper can be readily obtained from the general theory of polymer processing and polymer physics. The corresponding relationship of the samples in the diagram should be represented in the coordinates, rather than simply using different colors to guess in the legend. For example, figure 4 and Figure 5.

3)The results of this paper have some practical significance, and it is suggested to be revised and published.

Author Response

1) Different grades of polypropylene, due to the crystallinity of the structure of the difference, its own performance range is very large. Different recycled polypropylene may get different results and conclusions from the paper. The polypropylene and recycled polypropylene described in the paper are to be used with a strictly specified source and scope of collection.

This statement is true. The issue here is that such information is not published in data sheets and this information is therefore not available to us, even though it is of high relevance. We added in ”3 Materials and Methods” a clarifying sentence.  In end of “4. Conclusions” we added also a clarifying sentence. Unfortunately, this issue cannot be fully healed and this is weakness of our and similar studies.

2)The conclusions presented in this paper can be readily obtained from the general theory of polymer processing and polymer physics.

This is true. We added a clarifying sentence: “As expected from and in harmony with the general theory of polymer processing and polymer physics,“. We confirmed what could be expected. We see a value in this even though the novelty is restricted.

The corresponding relationship of the samples in the diagram should be represented in the coordinates, rather than simply using different colors to guess in the legend. For example, figure 4 and Figure 5.

We amended the diagrams.

3)The results of this paper have some practical significance, and it is suggested to be revised and published.

We amended the manuscript.

Reviewer 2 Report

The paper presents a study of the properties of virgin and mechanically recycled polypropylene.  There is a desire to recycle a greater proportion of plastics as part of the circular economy and to reduce waste going to landfill.  Therefore, understanding the properties of recycled materials and their potential applications is an important topic for investigation.  The present study collects secondary data from a range of recycling companies, and analyses them to study trends in the recycled vs fresh materials.  I believe the paper has some merits, but some concerns should be addressed before it could be deemed suitable for publication.

  1. The wide spread of results can indicate very different material properties, as well as potentially different testing standards, although it is suggested these are compliant with ASTMD638 and ISO527. Can the authors comment further on the reliability of the data from such a range of different sources?  In my opinion the paper might be stronger if different plastics had been obtained and subjected to the same new laboratory tests to determine their tensile properties.
  2. There seems to be some assumptions about correlation which are not proved by the data, for example in the conclusions it is stated ‘Even though it is impossible from the available market data to draw a direct dependency between the properties of virgin and recycled PP it can be assumed both correlate with each other.’ If it wasn’t possible to show correlation from the data why would it be safe to make this assumption?
  3. A sentence of the abstract does not seem to make sense ‘Even though the direct properties if recyclates could not brought into dependence to the properties of the virgin polymers from which the recyclates were made and not to the recycling parameters, the effect of recycling on the properties of PP became apparent as a result of this comparison.’ Consider rewording. 
  4. The figures are comprised of mainly Excel box plots. Although there is some explanation on page 9 of what these show, it is not clear from the figure legend and caption, what the data points, lines and shaded boxes represent.  I suggest a legend is added.  Additionally these figures seem to take up a lot of space given the amount of data represented.
  5. The discussion is somewhat limited, although gives some insight into the types of applications the recycled polymers could be suitable for, it could perhaps be enhanced and strengthened.

Author Response

The paper presents a study of the properties of virgin and mechanically recycled polypropylene.  There is a desire to recycle a greater proportion of plastics as part of the circular economy and to reduce waste going to landfill. Therefore, understanding the properties of recycled materials and their potential applications is an important topic for investigation. The present study collects secondary data from a range of recycling companies, and analyses them to study trends in the recycled vs fresh materials.  I believe the paper has some merits, but some concerns should be addressed before it could be deemed suitable for publication.

                We amended the document and tried to address concerns.

  1. The wide spread of results can indicate very different material properties, as well as potentially different testing standards, although it is suggested these are compliant with ASTMD638 and ISO527. Can the authors comment further on the reliability of the data from such a range of different sources? In my opinion the paper might be stronger if different plastics had been obtained and subjected to the same new laboratory tests to determine their tensile properties.

This objection is true and we agree. We are not able and it not within the scope of the study to purchase and test materials. That would be a different study. Such experimental study would hardly allow evaluating a broad range of data, as it is the case here. Information about reliability and specific test parameters were not available. We propose a study we physical experiments for a separate, experimental study. We wrote in “3 Materials and Methods” a clarifying sentences: “The data for V-PP lacked information on the molecular weight distribution, conformation, crystallinity, extrusion process, flexural strength, additives and the material's chemical composition. The data for R-PP lacked in addition information about the number of extru-sions and impurities. There was also no information about the grade of PP (ho-mo-polymer, random-copolymer, or block-copolymer).” Herby, we reveal and addresses weakness of our study and we provide readers the opportunity to consider them.

  1. There seems to be some assumptions about correlation which are not proved by the data, for example in the conclusions it is stated ‘Even though it is impossible from the available market data to draw a direct dependency between the properties of virgin and recycled PP it can be assumed both correlate with each other.’ If it wasn’t possible to show correlation from the data why would it be safe to make this assumption?

We weakened the sentence in the abstract and referred in the conclusion to chapter 1.2. Polymer degradation during recycling is well known. We did not have information about recyclates history. Multiple extrusion, what happens at recycling causes polymer degradation. Since we lack information about polymer recyclate history we decided to write the sentence weak and clearly state it is an assumption, we think a justified one.

  1. A sentence of the abstract does not seem to make sense ‘Even though the direct properties if recyclates could not brought into dependence to the properties of the virgin polymers from which the recyclates were made and not to the recycling parameters, the effect of recycling on the properties of PP became apparent as a result of this comparison.’ Consider rewording.

Please see 2.

  1. The figures are comprised of mainly Excel box plots. Although there is some explanation on page 9 of what these show, it is not clear from the figure legend and caption, what the data points, lines and shaded boxes represent. I suggest a legend is added. 

We agree that legend would make sense. However, we decided against a legend since box plots are well explained in databases such as “Wikipedia” and basic statistic books and basic information is very easily available via search programs such as google.

Additionally these figures seem to take up a lot of space given the amount of data represented.

That is right. We assume MDPI will do the editing and adaption of Figure sizes.

  1. The discussion is somewhat limited, although gives some insight into the types of applications the recycled polymers could be suitable for, it could perhaps be enhanced and strengthened.

We strengthened the discussion and added several additional references. To avoid overinterpretation and even speculation we limited our discussion.

Reviewer 3 Report

This study compared the properties of mechanically recycled PP (R-PP) and virgin PP (V-PP) by using data from recyclers, virgin plastic suppliers and literature. It is valuable for waste reutilization and resource recycling. I recommend it is published in RECYCLING.

Author Response

This study compared the properties of mechanically recycled PP (R-PP) and virgin PP (V-PP) by using data from recyclers, virgin plastic suppliers and literature. It is valuable for waste reutilization and resource recycling. I recommend it is published in RECYCLING.

Thank you for reviewing.

Round 2

Reviewer 2 Report

The authors have responded to all the comments and partially improved their paper.  I am happy to recommend it for acceptance.